# A Sol-Gel/Solvothermal Synthetic Approach to Titania Nanoparticles for Raman Thermometry

**DOI:** 10.3390/s23052596

**Published:** 2023-02-26

**Authors:** Thomas Pretto, Marina Franca, Veronica Zani, Silvia Gross, Danilo Pedron, Roberto Pilot, Raffaella Signorini

**Affiliations:** 1Department of Chemical Science, University of Padova, Via Marzolo 1, I-35131 Padova, Italy; 2Consorzio Interuniversitario Nazionale per la Scienza e Tecnologia dei Materiali (INSTM), Via G. Giusti 9, I-50121 Firenze, Italy; 3Institute for Chemical Technology and Polymer Chemistry (ITCP), Karlsruhe Institute of Technology (KIT), 76131 Karlsruhe, Germany

**Keywords:** temperature, nanothermometer, Raman, non-contact technique, anatase, nanoparticles, green synthesis

## Abstract

The accurate determination of the local temperature is one of the most important challenges in the field of nanotechnology and nanomedicine. For this purpose, different techniques and materials have been extensively studied in order to identify both the best-performing materials and the techniques with greatest sensitivity. In this study, the Raman technique was exploited for the determination of the local temperature as a non-contact technique and titania nanoparticles (NPs) were tested as nanothermometer Raman active material. Biocompatible titania NPs were synthesized following a combination of sol-gel and solvothermal green synthesis approaches, with the aim of obtaining pure anatase samples. In particular, the optimization of three different synthesis protocols allowed materials to be obtained with well-defined crystallite dimensions and good control over the final morphology and dispersibility. TiO_2_ powders were characterized by X-ray diffraction (XRD) analyses and room-temperature Raman measurements, to confirm that the synthesized samples were single-phase anatase titania, and using SEM measurements, which clearly showed the nanometric dimension of the NPs. Stokes and anti-Stokes Raman measurements were collected, with the excitation laser at 514.5 nm (CW Ar/Kr ion laser), in the temperature range of 293–323 K, a range of interest for biological applications. The power of the laser was carefully chosen in order to avoid possible heating due to the laser irradiation. The data support the possibility of evaluating the local temperature and show that TiO_2_ NPs possess high sensitivity and low uncertainty in the range of a few degrees as a Raman nanothermometer material.

## 1. Introduction

In a wide range of cases, from microelectronics and nanoelectronics to nanomedicine, the precise determination of the local temperature at the nanometer scale is essential to achieve control of the physical, chemical and biological processes involved. Large experimental initiatives are underway in the field of nanothermometry [1,2,3], from the generation of new materials, to the implementation of analytical techniques and prototypical tests of real devices, with the final aim of obtaining high sensitivity and good spatial resolution in the detection of the local temperature over a wide range of temperatures, in particular that of biological interest (293–323 K). The achievement of these objectives is the driving force for the generation of new materials to be used in nanothermometry. The determination of the local temperature of microscopic or nanoscopic objects can be carried out using different techniques [1,2,4], which are divided into contact techniques, mainly based on scanning microscopes [5,6,7], and contactless techniques, which exploit optical properties, such as luminescence [8,9,10] or diffusion [11,12]. Interesting features in the exploitation of Raman spectroscopy as a thermometric technique are the high spatial resolution, of the order of the diffraction limit of the laser probe (<1 μm), and good temperature resolution (1–10 K) over a wide wavelength range. Moreover, Raman is a non-destructive technique that can be applied to a wide range of samples and experimental conditions—it does not require complex sample preparation, requires small volumes of material, and can be applied in a wide range of environments, even during the course of chemical reactions or under extreme conditions of pressure and temperature [1,2,13].

The temperature can be measured from Raman spectra by determining the degree of shift in the position of a defined peak at different temperatures, by evaluating the broadening of its linewidth, or by measuring the peak intensity ratio of the anti-Stokes signal to the Stokes signal [12].

Nanoparticles’ unique characteristics have been extensively explored for use in the nanothermometry field to detect the local temperature [2]. The synthesis of nanoparticles with an appropriate and homogeneous size, structure, and morphology is an essential purpose to be achieved. In particular, the particle size and distribution are fundamental for their use in the biomedical field [14,15,16,17,18]. For practical applications, nanoparticles and nanostructures ranging from 5–10 nm [19,20] to 100–600 nm [21,22,23,24] have been tested.

Titanium dioxide, particularly the anatase polymorph, possesses all the characteristics that define a suitable material for Raman nanothermometry—it has a large Raman scattering cross-section, an intense and clearly distinguishable peak at low Raman shifts, and low absorbance at the wavelengths of excitation in the visible and NIR, which limits the heating of the sample due to the laser source. Moreover, it also has interesting chemical characteristics, such as high chemical stability and non-toxicity in biological environments. Lastly, commercial TiO_2_ NPs have already shown good properties as Raman active nanothermometers in the visible range [25,26].

The possibility of synthesizing pure anatase nanoparticles of well-defined dimensions and suitably functionalized to increase their solubility in biological media would open the way to the use of this material in the biomedical field. In the literature, numerous sol-gel syntheses of titania have been reported; however, they often require a calcination step at temperatures above 500 °C. Some articles report sol-gel methods combined with the use of ultrasound treatments [27] or the use of reflux conditions at 70 °C [28], while others adopt solvo/hydrothermal treatments, taking advantage of mild reaction conditions [29]. Sol-gel and solvothermal syntheses, both based on mild synthesis conditions, allow controlling of the shape, size, structure, and composition of NPs by optimizing parameters, such as the temperature, reaction time, and the amount and ratio of reactants, additives, and ligands.

In this work, titanium dioxide NPs, as an anatase polymorph, were synthesized, employing an optimized combination of sol-gel and solvothermal synthetic approaches to obtain samples of pure anatase, with defined crystallite dimensions and a bipyramidal morphology characterized by a high Raman signal. The synthesis process of anatase nanoparticles was optimized and fine-tuned in compliance with sustainability criteria from a green perspective (i.e., low temperature, non-hazardous solvents and chemicals). The TiO_2_ NPs were characterized by SEM, XRD and Raman techniques to investigate their morphology and microstructure, respectively. In particular, their nanothermometer properties were evaluated in the visible region at 514.5 nm by anti-Stokes (aS) and Stokes (S) Raman measurements collected in the 293–323 K range.

## 2. Materials and Methods

### 2.1. Synthesis of TiO_2_ NPs

The process proposed for the synthesis of anatase NP consists of the combination of a sol-gel reaction with a hydrothermal treatment performed within a hydrothermal reactor, as schematically represented in Figure 1.

In all the syntheses, the titanium alkoxide precursor was titanium (IV) tetraisopropoxide ($\mathrm{Ti}{\left(O^{i}\mathrm{Pr}\right)}_{4}$ or TTIP), chosen for its low toxicity. The most relevant parameters in this process are the hydrolysis ratio, $r=\frac{mol\;\left(H_{2}O\right)}{mol\;\left(\mathrm{Ti}{\left(O^{i}\mathrm{Pr}\right)}_{4}\right)}$, the pH (acid/base catalysis and surface stabilization of NPs) and the reaction environment, including the type of solvent and ionic strength, the absolute and relative concentrations of reagents, and the hydrothermal treatment time. These parameters were systematically optimized; three routes were identified as the best-performing ones.

The three synthesis strategies adopted, summarized in Table 1, differed in the choice of the following reaction conditions: pH, composition of the reaction mixture, and possible use of agents for the morphological control of the material: the peptizing agent promotes the dispersion of the nanoparticles by introducing, as example, hydrogen bonds on the hydroxylated surface of the NPs [30].

#### 2.1.1. Synthesis with Ethylene Glycol

The synthesis used a water/ethylene glycol mixture of variable composition and tetraethylammonium hydroxide (TEAOH) as a peptizing base. Relative amounts of diol and peptizing agent were expressed as a function of moles of TTIP. The molar ratio (mole ethylene glycol/mole TTIP) = 60 was kept constant in all reactions in this series to guarantee the same stabilization and dispersion of the synthesized NPs.

The composition of the reagent mixture was varied (as more extensively described in the Supporting Materials), by increasing the hydrolysis ratio and varying the quantity of the TEAOH peptizing base. Furthermore, the temperature and time of the hydrothermal treatment were varied to verify the effect on the resulting titania properties.

Ti(O^i^Pr)_4_ (97% by weight in isopropanol, Merck 546-68-9, Merck, Milano, Italy) was dissolved in water and ethylene glycol (99% pure, Carlo Erba, Milano, Italy) and TEAOH (35% by weight in water, Sigma-Aldrich, Merck, Milano-Italy), using the following molar ratios: mol(H_2_O)/mol(Ti(O^i^Pr)_4_ = min, 7, 68, 75, 100, 125, 150, 175, 346, 708 and 1431 (where min is the minimum amount of H_2_O contained in the 35% TEAOH solution) and mol(TEAOH)/mol(Ti(O^i^Pr)_4_) = 0.1. 0.22 and 4.5. After 30–40 min of magnetic stirring at room temperature, the solution was transferred to a Parr stainless steel hydrothermal bomb (Model 4745 General Purpose Acid Digestion Vessel, 23 mL volume, temperature limit of use 150 °C, held up to 483 bar and Model Berghor, 50 mL volume, temperature limit of use 250 °C, 330 bar), which was then heated at 150 or 180 °C for 5, 12 or 24 h. The working syntheses led to the formation of a white precipitate which was isolated by centrifugation and washed with deionized water and ethanol. To obtain the final powder, the precipitate was dried in a dryer with silica gel and placed under vacuum for one night.

#### 2.1.2. Acid Synthesis

Titanium [IV] tetra-isopropoxide (Ti(O^i^Pr) 4, 97% by weight in isopropanol, Merck 546-68-9, KGaA, St. Louis, MO, USA) was dissolved in water and acetic acid (glacial, ACROS, hermo Fisher Scientific, Geel, Belgium) was added. The following molar ratios were used (see also Appendix A): mol(CH_3_COOH)/mol(Ti(O^i^Pr)_4_ = 105, 210 and mol(H_2_O)/mol(Ti(O^i^Pr)_4_ = 1.66, 3.32, 4.15, 10.4, 16.6, 41.5, 75, 125, 150, 225, 346, 708, 1431 (the starting values were taken from [31]; they were then varied to check the effect). The procedure adopted was similar to that for glycol synthesis but with a hydrothermal treatment temperature of 150 °C for 24 h.

#### 2.1.3. Basic Synthesis

These syntheses were performed at a buffered pH 9.5 obtained by dissolving an ammonium salt (NH_4_Cl, >99.5% pure Sigma-Aldrich, Merck, Milano-Italy, or (NH_4_)_2_SO_4_, >99% pure, Baker, Thermo Fisher Scientific, Waltham, MA, USA) in an ammonia solution (NH_4_OH) 28% by weight in water, Sigma-Aldrich, Merk, Milano, Italy); the salt used to create the buffer solution was varied to check whether the anion of the ammonium salt had an effect on NP extraction, on the polymorph obtained and on morphology. The following hydrolysis ratios (*r*) were used (see also Appendix A for details): mol(H_2_O)/mol(Ti(O^i^Pr)_4_ = 16.6, 41.5, 75, 128.7, 164.9, 225.2, 345.9, 707.9, 1431.8. The hydrothermal treatment was performed for 24 h at 150 or 180 °C.

### 2.2. SEM and XRD Characterization of TiO_2_ NPs

SEM analyses were performed with a Zeiss Sigma (Oberkochen, Germania) HD microscope, equipped with a Schottky FEG source, one detector for backscattered electrons, and two detectors for secondary electrons (InLens and Everhart Thornley). Topographic measurements were performed at 20 kV. X-ray diffractometry (XRD) was used to analyze the crystal structure and the crystallite size of NPs. The diffractograms of the samples were acquired with a Bruker D8 Advance diffractometer, using a CuK_α_ (λ = 1.5406 Å) radiation source. The range was 2θ = 20–80°, with a scan step of 0.026° 2θ, and a 0.3 s acquisition per step. The crystallographic phase identification was performed using a search and match procedure, using Bruker Diffrac EVA software (V5.1).

Titanium dioxide was used in powder form, as just synthesized, for the SEM and XRD analyses.

### 2.3. Raman Characterization of TiO_2_ NPs

The micro-Raman set-up was equipped with an Ar^+^ laser providing a 514.5 nm wavelength (Spectra Physics, Stabilite 2017), with an output power of 1 W. The back-scattered Raman signal, separated from the Rayleigh scattering by an edge filter, was analyzed with a 320 mm focal length imaging spectrograph and a liquid-nitrogen-cooled CCD camera. Each Raman spectrum was obtained through 10 acquisitions of 10 s (10 a × 10 s), using a laser power of 0.08 mW incident on the sample, and a spectrometer entrance slit of 110 µm. The Raman spectra were collected in different regions of the samples.

Titanium dioxide was characterized as a powder pressed on a KBr pellet sample. A finely ground powder spatula tip was pressed at 4 bar for 5 min on KBr tablets; it possessed a final thickness of a few hundred μm.

### 2.4. Raman Nanothermometry

The micro-Raman setup used for the nanothermometry measurements was equipped with an Ar^+^/Kr^+^ gas laser (Coherent, Innova 70, Santa Clara, CA, USA), providing the line at 514.5 nm. The laser beam was coupled to a microscope (Olympus BX 40, Tokyo, Japan) and focused on the sample with 20× objectives (Olympus SLMPL). Raman scattering was coupled into the slit of a three-stage subtractive spectrograph made up of a double monochromator (Jobin Yvon, DHR 320, Horiba, Kyoto, Japan), working as a tunable filter that rejected elastic scattering, and a spectrograph (Jobin Yvon, HR 640). The Raman signal was detected by a liquid-nitrogen-cooled CCD. The Raman instrument was interfaced with a temperature control stage (Linkam, THMS600/720, Tadworth, UK), used to change the sample temperature in the range of 77–600 K. The sample was uniformly heated/cooled to reach the desired temperature at a rate of 2 K/min and a thermalization time of at least 15 min. Once the thermalization process was completed, consecutive Stokes and anti-Stokes measurements were conducted and repeated to obtain a consistent set of data to calculate the local temperature of the sample. For the temperature measurements, a power intensity equal to or less than 1 mW was used to avoid laser-induced heating. Titanium dioxide was inserted in the temperature control stage as a powder pressed on a KBr pellet sample.

#### Local Temperature Determination

The temperature can be determined from the ratio between the intensities of two homologous Stokes and anti-Stokes Raman peaks of a given vibrational Raman active transition using the following equation:(1)$$\frac{I_{aS}}{I_{S}}=A{\left(\frac{\nu_{0}+\nu_{m}}{\nu_{0}-\nu_{m}}\right)}^{3}e^{-\frac{h\nu_{m}}{k_{B}T}},$$
where $\nu_{0}$ represents the laser frequency, $\nu_{m}$ is the frequency of the Raman active vibrational mode, h is Planck’s constant, $k_{B}\;$ is the Boltzmann constant and A represents a constant that is mainly determined by instrumental factors [12]. The thermal sensitivity (*s*) of the response in the temperature range is the first derivative of Equation (1) as a function of the temperature T:(2)$$\partial \frac{I_{aS}}{I_{S}}/\partial T=\left|-{\left(\frac{\nu_{0}+\nu_{m}}{\nu_{0}-\nu_{m}}\right)}^{3}\frac{h\nu_{m}}{k_{B}T^{2}}e^{-\frac{h\nu_{m}}{k_{B}T}}\right|,$$

It decreases as the temperature increases and depends on the frequency of the Raman mode; a low frequency mode is more sensitive [26]. The temperature resolution ($\Delta T_{min}$), is given by the ratio between the uncertainty in the variation of the signal and the sensitivity of the measurement:(3)$$\Delta T_{min}=\frac{\sigma }{s}$$
where the uncertainty is expressed as the standard deviation (*σ*).

For the determination of the local temperature, Raman Stokes and anti-Stokes Raman spectra of anatase were collected in the temperature range between 293 and 323 K, with increments of 5.0 K, interspersed with 15 min of thermalization. The aS and S branches were acquired sequentially by moving the gratings of the triple spectrograph; three pairs of Raman spectra (Stokes and anti-Stokes) were acquired at each temperature.

## 3. Results

All the synthetized samples were structurally, morphologically and optically characterized to choose the best-performing ones, in terms of size, crystallinity and Raman signal intensity. These were used as test materials for the temperature measurements.

An example of the complete characterization of the sample TP22.10, prepared with acid synthesis with *r* = 3.32, $\frac{mol\;\left({\mathrm{CH}}_{3}\mathrm{COOH}\right)}{mol\;\left(\mathrm{Ti}{\left(\mathrm{OiPr}\right)}_{4}\right)}\;$= 105, at 150 °C for 24 h, is presented in Figure 2 and Figure 3.

From the SEM images, it was possible to observe the morphological characteristic of the nanocrystals, which depend on the synthesis conditions used. The elongated and tapered nanostructures were the result of specific growth mechanisms under hydrothermal conditions. The image in Figure 2b shows a single small particle with a nanorod shape, with a major axis around 700 nm and a minor one around 300 nm.

The XRD pattern of the sample TP22.10 showed the presence of a pure anatase phase in the sample: no spurious phases were found in the XRD patterns, as reported in Figure 3a. The diffraction peaks were indexed with powder diffraction standard data using the Crystallography Open Database. The crystallite size of TiO_2_ NPs was estimated from the broadening of the anatase (101) reflection, deconvoluted with a Lorentzian function to determine the full width at half maximum (FWHM). Using the Debye formula [32], the calculated crystallite size for the TP22.10 nanoparticles was found to be 13 nm.

The Stokes Raman spectrum of the TP22.10 sample recorded at room temperature is reported in Figure 3b. The spectrum shows an intense peak centered at 147 cm^−1^, with a shoulder at 197 cm^−1^, and three peaks at 397, 515 and 640 cm^−1^ with lower intensity, which are comparable to data reported in the literature [33]. The corresponding Raman active modes are also indicated in the figure. The uniformity, in terms of the positions and intensity of the Raman signal acquired at the various positions of the sample, clearly indicates the homogeneity of the synthesized sample tested over a wide spatial range.

The more interesting results, with ethylene glycol synthesis, were obtained at 150 °C. They are summarized in Table 2. The optimal molar ratio of 1,2-ethanediol:TEAOH was established to be 60:4.5; the hydrolysis ratio was increased from a minimum value of 68.5, corresponding to the water present in the 35% (by weight in water) TEAOH solution, to 1431. Only using these conditions, it has been possible to obtain a powder sample. At low hydrolysis ratio values, the samples were observed to be amorphous (TP23.1 and TP23.2), while in the hydrolysis ratio range of 100–175, anatase was obtained as the only pure phase. When the ratio was further increased, the formation of an anatase/rutile mixture (TP26.4) and rutile (TP26.5) was observed.

The parameters obtained from the fitting of the E_g(1)_ Raman mode in the spectra, such as the Raman intensity normalized for the excitation input power, the peak position and FWHM, are also reported in Table 2 and shown in Figure 4, together with the results of the XRD, in terms of the crystallite size.

Figure 4a shows that, as the hydrolysis ratio increased in the range 100–175, the crystallite diameter decreased by approximately 50%, from 49 nm large crystallites (TP23.3, *r* = 100) to 26 nm crystallites (TP23.6, *r* = 175). This can likely be ascribed to a faster hydrolysis and condensation process, leading to the rapid nucleation of the NPs [34,35,36]. Furthermore, the size of the crystallite did not seem to be correlated with the intensity of the Raman peak E_g(1)_: the samples had high intensities, with the exception of the sample synthesized with *r* = 100 and *r* = 708 and 1450. The width at half-height (FWHM) of the Raman peak decreased only at a high hydrolysis ratio due to the presence of a rutile phase together with anatase. The center of the peak moved to higher Raman shifts.

As can be understood by comparing the SEM images reported in Figure 5 and the Appendix A with the Raman data, the most intense Raman signals were obtained when the nanoparticles were of homogeneous morphology (TP23.6, TP9), or when ellipsoidal elongated particles were obtained, or when particles appeared to have fused together to form rods (TP23.4). Nanostructures with dimensions ranging from 30 to 300 nm were obtained with glycol synthesis.

The acid synthesis always led to the formation of crystalline anatase, and only in a few cases (outlined in the Appendix A) did the XRD patterns indicate the co-presence also of an amorphous component in such amounts as not to visibly alter the Raman properties of the material. The results of the characterizations of samples prepared with the acid synthesis are shown in Figure 6. In the images, the correlation diagrams were drawn between the diameter of the crystallite, the position, intensity, width at half height of the Raman peak E_g(1)_ of the anatase, and the hydrolysis ratio used. The diagrams also show the characterization data of the samples synthesized twice, with *r* = 3.32 or with *r* = 16.6, to evaluate the reproducibility of the synthesis.

The crystallite size and the intensity of the Raman signal decreased as *r* increased, while both the shift and the amplitude of the Raman peak E_g(1)_ decreased with increasing hydrolysis ratio. We can easily explain the decrease in crystallite size as a function of the hydrolysis ratio because the hydrolysis rate was increased using a larger amount of water in the reaction mixture, producing smaller crystallites. On the other side, the Raman peak position and the width behavior were different from what has been observed in the literature, as they shifted at lower values with increasing hydrolysis ratio, i.e., with decreasing crystallite size. This discrepancy may be attributed to the presence of structural defects in the material, which may have broadened the diffraction peak and consequently the crystallite size [37,38]. The highest intensities of the peak E_g(1)_ were obtained at low *r* values and a low hydrolysis ratio.

The crystallite dimensions estimated from the XRD measurements were between 10 and 30 nm; with this synthesis, crystallites with dimensions smaller than the ones obtained with the glycol synthesis were obtained, indicating a lower size limit of the NPs. Moreover, the SEM images, reported in Figure 7 and Appendix A, confirmed the capability of tuning and reducing the dimension of the nanostructures obtained by changing the experimental parameters—the morphological control of the material with the acid synthesis seemed to depend substantially on the hydrolysis ratio.

At higher dilutions (*r* ≤ 4.15, TP22.10) the formation of larger nanoparticles was favored, with hydrolysis and condensation occurring more slowly in this case; a slow nucleation of a few nuclei is hypothesized to occur with different sizes incorporating other nuclei during the hydrothermal treatment. The results obtained in terms of shape and size were similar to those obtained with the glycol synthesis.

At small dilutions (*r* = 150, TP25.5), a fast nucleation of numerous nuclei, with a narrow size distribution, followed by slow growth during the hydrothermal phase, was probably obtained; faceted morphologies were obtained. When the hydrolysis ratio was further increased, more aggregated nanoparticles smaller than 20 nm were obtained; micrometer sized aggregates could also be seen.

Regarding the properties of the Raman signal, its variability seemed to be linked to the diameter of the crystallites; the more intense signals (around 200,000 and 300,000 counts/mW) were obtained with low *r* values (in between 1.66 and 3.74), while a good signal was observed in all the other conditions (see Appendix A for details). The maximum variation of the position of the E_g(1)_ peak and of its FWHM was 3 and 4 cm^−1^, respectively, reflecting the presence of nanoparticles with different dimensions.

The XRD patterns and Raman spectra of the samples synthetized under basic conditions also revealed the presence of brookite in a negligible amount (0.04–0.07% wt) with respect to anatase. The presence of these traces did not influence the Raman E_g(1)_ peak of anatase, and was, therefore, not a problem for thermometry purposes.

The correlation diagrams for the basic synthesis are reported in Figure 8. Over the whole range of the hydrolysis ratios, the basic synthesis led to more reproducible results: the size estimate of the crystallites was always around 20 nm.

For this synthesis, the properties of the Raman signal varied little as *r* varied; the maximum variation in the position of the E_g(1)_ peak and of its FWHM were 0.6 cm^−1^ and 1.1 cm^−1^, respectively.

The SEM images of these samples (TP21.1 is reported in Figure 9a as an example) showed nanoparticle aggregates of faceted shape with a narrow size distribution centered around the values of 25–50 nm. The basic synthesis seems not to have been affected by the variation in the hydrolysis ratio.

The signal intensities were, on average, lower, but comparable with those obtained from the other syntheses. As can be seen from the diagrams in Figure 10, the intensity values also varied little as a function of both the diameter of the crystallites and throughout the range of variation of the hydrolysis ratio. This effect may be attributed to the crystallinity of the sample obtained [39].

A comparison between the samples obtained with the different syntheses is shown in Figure 10, where the intensity of the Raman peak is reported both as a function of the size of the crystallites (a) and as a function of the hydrolysis ratio. The acid synthesis allowed modulation of the dimensions of the crystallites and of the NPs together with the intensity of the Raman signal, while the synthesis with glycol allowed modulation of the dimensions of the crystallites, but not of the Raman intensities, and the basic one maintained a constancy in the two parameters.

The determination of the local temperature was obtained from Stokes and anti-Stokes Raman spectra of anatase collected over a defined temperature range on selected samples. From the first Raman characterization performed on all the synthetized materials, a set of samples prepared with the three different syntheses was selected on the basis of the most intense and homogeneous Raman signal to be exploited for nanothermometry. These samples were TP9 for glycol synthesis, TP22.10 and TP25.1 for acid synthesis, and TP21.3 and TP26.9 for basic synthesis (see Appendix A for details on intensity).

Figure 11a reports the Stokes and anti-Stokes Raman spectra of the E_g(1)_ peak of TP9 sample collected in the temperature range between 293.2 and 323.2 K. All these peaks were interpolated with a Lorentzian function to determine their area and intensity ($I_{aS}$ and $I_{S}$). The values of the ratio between the peak intensities ($I_{aS}/I_{S}$), together with the corresponding absolute uncertainty, were obtained and are reported in Figure 11b as a function of the temperature. Equation (1) was used to fit these data as a function of the sample temperature, which has an almost linear trend in this interval, starting from the experimental parameters reported in Table 3.

The instrumental constant was considered as a free parameter of the fitting and determined through calculation iterations. Figure 11b shows the fitting curve, and Table 3 reports the value of the constant A and the relative sensitivities as a percentage.

The obtained value was in agreement with the one obtained in the literature using a commercial TiO_2_ powder [26], confirming the quality of the synthetized nanomaterial. The figure also reports the relative sensitivities, which decreased as the temperature increased [26,40]. The thermal resolution (absolute uncertainty in determining the temperature), as well as the percentage accuracy, varied according to the calibration temperature.

Regardless of the type of synthesis used to produce the anatase and the intensity of the Raman signal at room temperature, it can be seen that in all cases it was possible to obtain a nearly linear relationship between the calibration temperature and the intensity ratio aS/S. The graphical representation of the response curve of the nanothermometers tested, obtained by fitting the experimental data with Equation (1), is shown in Figure 12. It shows that the response data of the nanothermometers tested followed the same increasing trend with the calibration temperature: the intensity ratio grew as the probability of aS events increased with the temperature, with a similar speed in all cases.

The calibration procedure of the nanothermometers enabled obtaining the value of the calibration constant A, reported in Table 4, together with the characteristics of the material, such as the Raman peak position, XRD crystallite and SEM nanoparticle dimensions.

The difference in size between the crystallites, highlighted with XRD, and the nanostructures, clearly visible with the SEM images, was due to the fact that the NPs were formed by several crystallites.

All the tested materials performed well in the temperature range 298–323 K, and, for Raman thermometry purposes, had comparable performance with commercial anatase, the data for which were taken from [26]. This work confirmed that, at a laser power of 1 mW, there is no self-heating of the sample; for the Raman shift E_g(1)_ mode close to 143 cm^−1^, the calibration constant assumes values close to the reference value.

For the different nanothermometers synthesized, the relative error in determining the temperature was calculated using the formula:(4)$$err=\frac{\left|T_{measured}-T_{real}\right|}{T_{real}},$$
where *T_measured_* is the value obtained from the signal of the nanothermometer, while *T_real_* is given by the thermostat at the different calibration temperatures. The relative error, depicted in Figure 13a, can also be given in percentage terms, as reported in Figure 13b.

## 4. Discussion and Conclusions

The need for and importance of using biocompatible nanosystems which allow the determination of local temperature in the biomedical field is the main thrust of this research work. From this point of view, two important purposes should be carefully considered. The first is the control of the dimensions and the dispersity of the nanomaterial; only small dimensional structures, in the range of tens of nanometers, can be suitably incorporated into biological systems. The second is the concomitant maintenance of the optical properties in terms of the quality and intensity of the detectable signal. From these requirements the need arises to synthetize anatase nanoparticles with dimensions modulated in the nanoscopic range, which can be used as active Raman materials for the determination of the local temperature. Control over the size, morphology and optical properties can be achieved through bottom-up synthesis. A green method was adopted which was sustainable both from an energy point of view and from the point of view of the toxicity and hazard of the reagents. The materials were synthesized by a sol-gel process combined with hydrothermal treatment, following three different syntheses: the first using a mixture of ethylene glycol and water as solvent, with tetraethylammonium hydroxide as a peptizer; the second using an aqueous solution of acetic acid as a catalyst; the third using a buffered ammonia solution at pH 9.5. The precursor used in all three cases was titanium tetraisopropoxide, which is very reactive and has low toxicity.

All three syntheses, optimized in term of reaction time and hydrothermal treatment temperature, were effective in obtaining anatase of different dimensions by varying the hydrolysis ratio, a parameter that was accurately controlled within the same range of values. Nanostructures having dimensions ranging from tens of nm to hundreds of nm were obtained, as underlined also by the SEM measurements presented in Figure 14. The results of seven different samples are reported in the diagram of Figure 14 as examples: TP22.10 and TP25.5 samples for the acid synthesis, TP21.1 and TP21.4 for the basic synthesis and TP7, TP9 and TP23.6 for the glycol synthesis. Within each synthesis, two different samples made with different hydrolysis ratios are reported: around the lowest and the highest probed for the synthesis to highlight the size of the NPs obtained. For the synthesis with glycol, a sample synthesized at 180 °C is also reported (TP7), which is identical to its correspondent, synthesized at 150 °C (TP9). On each box of the diagram, the central line represents the median of the size distribution, and the bottom and top edges of the box indicate the 25th and 75th percentiles, respectively. The whiskers extend, instead, to the most extreme data points in the distribution that are not considered outliers. Adopting the synthesis in an acid environment, the morphology and the size were modulated, from tens of nm NPs to hundreds of nm NPs, by regulating the hydrolysis ratio. Concerning the basic synthesis, within the range of variability of the hydrolysis ratio, it was possible to obtain nanostructures of faceted morphology and constant size of around tens of nm. With the glycol synthesis, nanostructures with dimensions in the range of hundreds of nm were obtained.

The average size of the crystallites was calculated using the Scherrer formula. It can be seen that the materials synthesized with the basic synthesis were made up of smaller crystallites, of around 20 nm. This dimension reflects what could be observed from the SEM images—for the basic synthesis, the choice of the hydrolysis ratio seems to have had little influence on the nanoparticle dimensions and morphology, therefore it represents a more repeatable synthesis. The synthesis with glycol yielded materials with larger crystallites, of around 40 nm. The acid synthesis appeared to provide more control over the nanoparticle morphology than over the crystallite size.

The intensity of the Raman signal was highly modulated with the hydrolysis ratio of the acid synthesis, reaching more than one order of magnitude higher at a low hydrolysis ratio with respect to the higher. The Raman intensities of the samples obtained with the basic synthesis were comparable with those obtained with samples prepared with acid synthesis with high hydrolysis values. The samples prepared with glycol synthesis, on the other hand, had intensity values intermediate between the two extremes. In any case, the intensity values obtained were such as to allow their use for local temperature determination using anti-Stokes and Stokes measurements, as demonstrated with the measurements performed in the temperature range 293–323 K.

For the practical use of TiO_2_ nanoparticles in biological systems, such as cells, a narrow and centered size distribution at dimensions in the range 10–50 nm will allow direct NP uptake. This dimensional range is easily achieved with the basic synthesis, which has been proved also to be the most reproducible, being less affected by variation in the reaction conditions. It can also be obtained with the acid synthesis working at high values of the hydrolysis ratio. Larger NPs can still be useful after proper functionalization.

The characterization of the optical properties through Raman spectroscopy has highlighted, within a given type of synthesis, the materials with the best signal-to-noise ratio, as regards the E_g(1)_ peak characteristic of anatase. This signal underwent variations in the Raman shift and in the amplitude at half height, which, according to the literature, were ascribable to the size of the crystallites and to the presence of lattice defects of a compositional type or derived from mechanical or thermal stress. The record of the Stokes and anti-Stokes branches of the Raman spectrum enabled performing Raman nanothermometry through the relationship between the intensity ratio aS/S of a peak and the temperature. It was possible to calibrate the Raman thermometer at different temperatures between 293 and 323 K to obtain the calibration constant. The calibration constants, estimated on a selected set of samples, were, in all cases, comparable with those determined for a commercial anatase powder described in the literature. The performance of the nanothermometer was, therefore, comparable through the relative sensitivity, which decreased as the calibration temperature increased. The synthesis of anatase of different dimensions and morphology opens the way to the possibility of preparing materials with well-defined characteristics in terms of dimensions, for example, and surface functionalization, and is the first step for the creation of composite materials. Only by controlling the dimensions and suitably functionalizing the external surface of the nanostructures will it be possible to internalize and localize them in biological systems. Moreover, it will open the way to the preparation of composite materials in which the nanothermometer, anatase, can be combined, for example, with a plasmonic material capable of amplifying its thermometric sensitivity and, at the same time, functioning as a local nanoheater. Finally, the use of anatase, as a Raman active material, enables monitoring of the local temperature over a wide wavelength range, from the visible to the near-IR range.

An important point will be to study the nanothermometry performance of the synthesized nanoparticles, not in the form of powder pressed on KBr tablets, but directly in aqueous suspension. Preliminary tests have shown encouraging results—the Raman signal of the colloidal suspension is high enough to be recorded even with attenuated power. In this regard, it will be important to ensure the dispersion of the nanoparticles through an electrostatic barrier (pH-dependent): the nanoparticles synthesized at acid or basic pH were found to be less aggregated.

The results obtained are very encouraging in view of the use of titania NPs as local temperature probes in biological systems. The use of a biocompatible material, combined with the possibility of modulating the dimensions of the NPs between the tens and hundreds of nm, and the excellent Raman signal of the nanosystems, pave the way for their practical use. There are good prospects in the future for the use of these nanoparticles in the in vitro and/or in vivo study of local temperature within living tissues to investigate cellular metabolism with good thermal resolution and excellent spatial resolution.

## Figures and Tables

**Figure 1 sensors-23-02596-f001:**
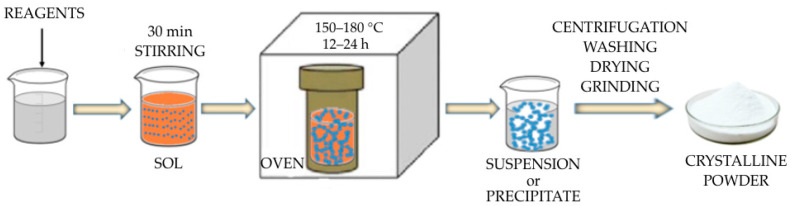
Scheme of the TiO_2_ NPs synthesis.

**Figure 2 sensors-23-02596-f002:**
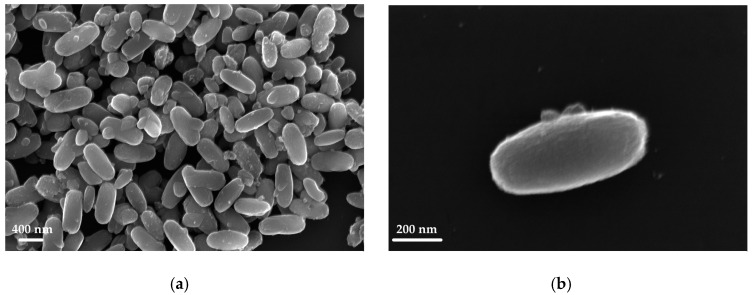
SEM image of the TP22.10 sample (**a**,**b**) zoomed in on a single small particle.

**Figure 3 sensors-23-02596-f003:**
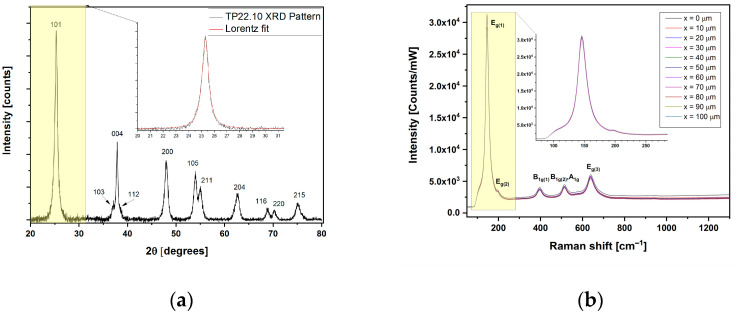
Characterizations of the TP22.10 sample (**a**) XRD pattern, with zoom in on 101 peak, and (**b**) Raman spectra collected at 514.5 nm at room temperature, at different positions of the TiO_2_ NPs sample, with zoom in on mode E_g(1)_, at 147 cm^−1^.

**Figure 4 sensors-23-02596-f004:**
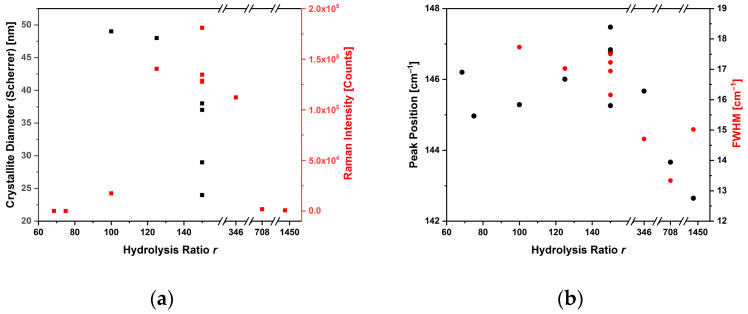
Plots correlating the crystallite diameter and intensity (**a**), position and width (**b**) of the Raman E_g(1)_ peak, with the hydrolysis ratio used in the anatase glycol synthesis.

**Figure 5 sensors-23-02596-f005:**
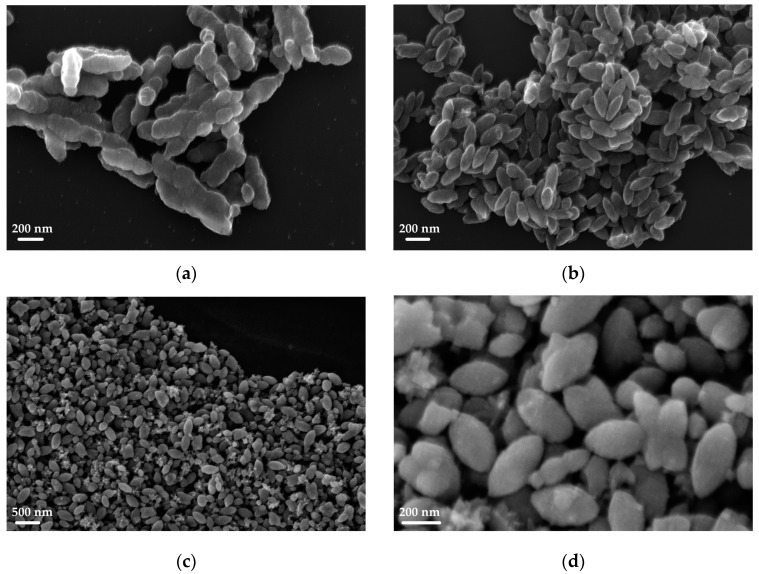
SEM images of TP23.4 (**a**), TP23.6 (**b**) and TP9 (**c**,**d**).

**Figure 6 sensors-23-02596-f006:**
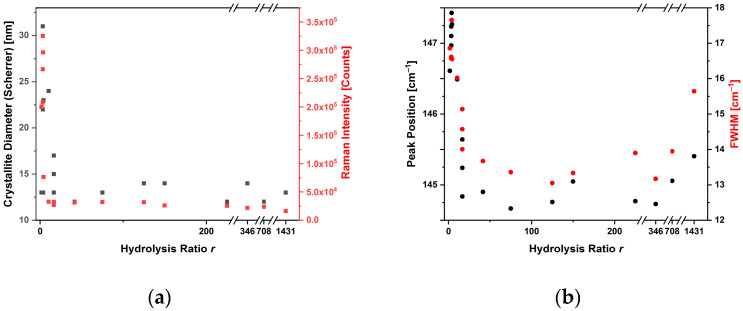
Plots correlating the crystallite diameter and intensity (**a**), position and width (**b**) of the Raman E_g(1)_ peak with the hydrolysis ratio used in the anatase acid synthesis.

**Figure 7 sensors-23-02596-f007:**
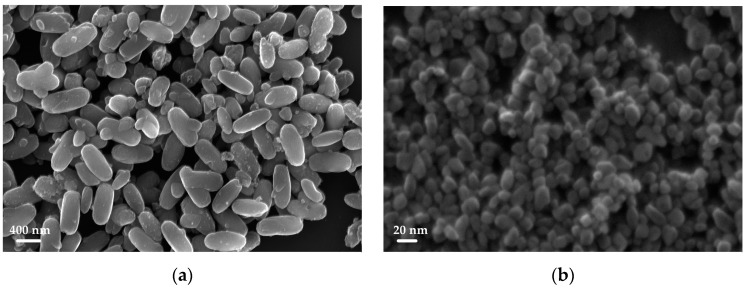
SEM images of TP22.10 (**a**) and TP25.5 (**b**).

**Figure 8 sensors-23-02596-f008:**
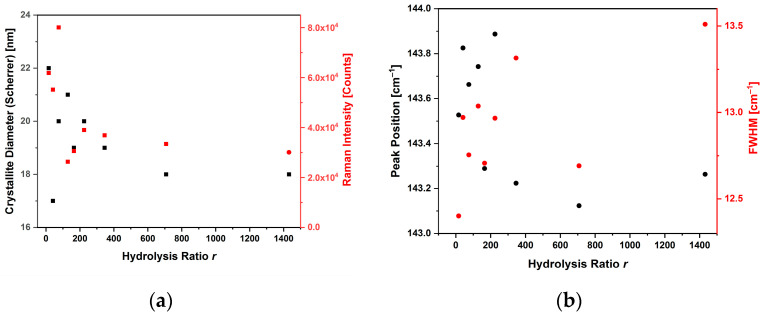
Plots correlating the crystallite diameter and intensity (**a**), position and width (**b**) of the Raman E_g(1)_ peak, with the hydrolysis ratio used in the anatase basic synthesis.

**Figure 9 sensors-23-02596-f009:**
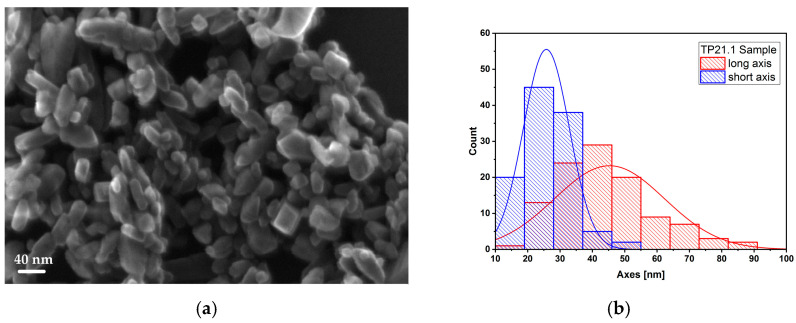
SEM image of TP21.1 sample (**a**) and histogram (**b**) representing the dimensional distribution of NPs.

**Figure 10 sensors-23-02596-f010:**
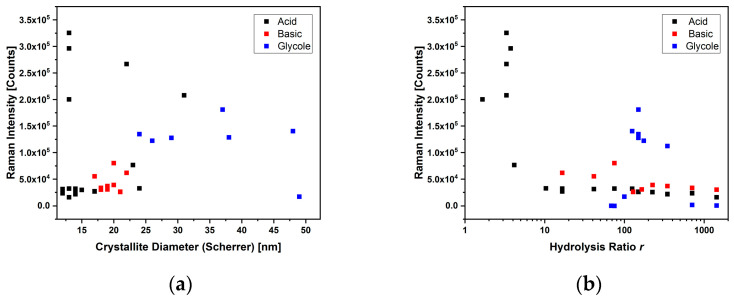
The plots correlate the Raman peak intensity with the crystallite diameter (**a**) and hydrolysis ratio (**b**) for all syntheses: acid (black squares), basic (red squares) and with glycol (blue squares).

**Figure 11 sensors-23-02596-f011:**
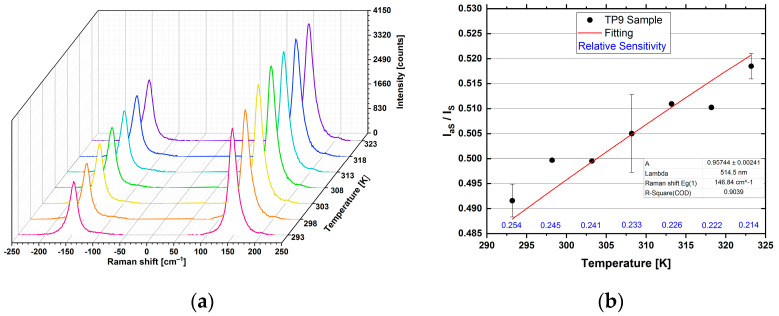
Thermometry carried out on sample TP9. Raman Stokes and anti-Stokes spectra acquired at 514.5 nm, at 1 mW and 1 s × 5 a (**a**). Calibration curve, through the connection of the data with the thermometric equation (**b**).

**Figure 12 sensors-23-02596-f012:**
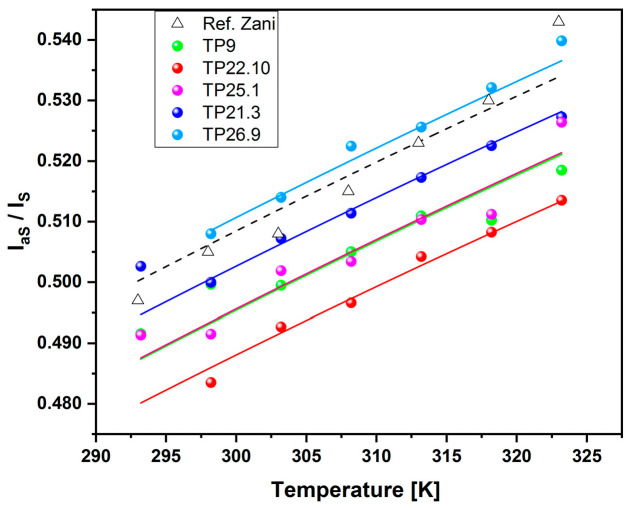
Response curves of the tested materials: TP9 sample (green dots), TP22.10 (red dots), TP25.1 (violet dots), TP21.3 (blue dots) and TP26.9 (sky blue dots). Literature data, open triangles, are also reported for comparison [26]. The error bars are omitted to make the graph easier to see.

**Figure 13 sensors-23-02596-f013:**
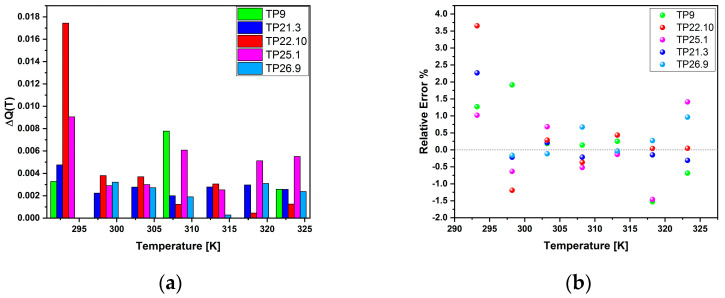
Comparison diagram between the relative (**a**) and percentage (**b**) errors committed by the different thermometers at the calibration temperatures.

**Figure 14 sensors-23-02596-f014:**
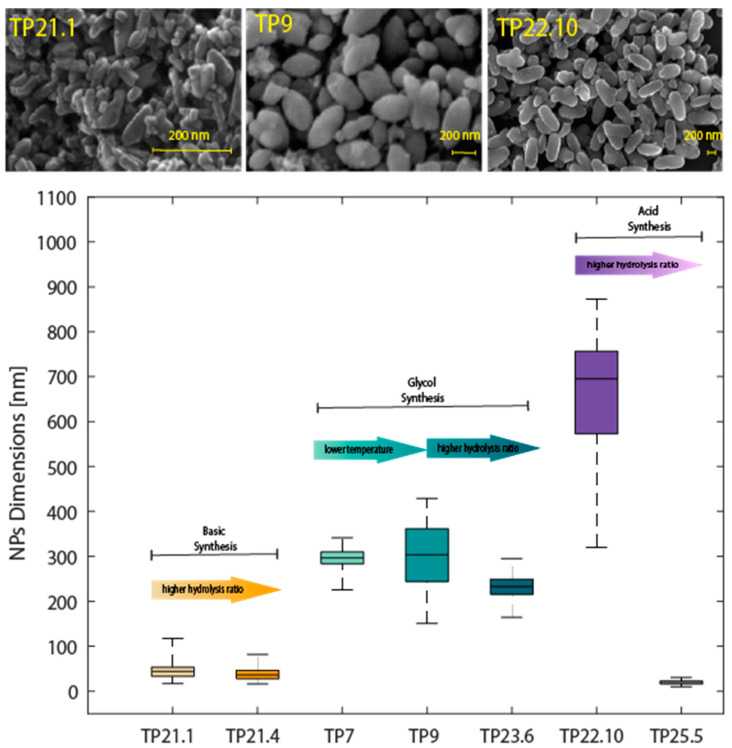
Diagram correlating the different syntheses and the NP dimensions. The morphology of some samples is also reported.

**Table 1 sensors-23-02596-t001:** Types of syntheses used for the preparation of anatase NPs.

Synthesis	Solvent	pH	Peptizing Agent
**Ethylene Glycol**	Ethylene glycol/H_2_O	Basic	TEAOH
**Acid**	Acetic acid/H_2_O	Acid	CH_3_COOH
**Basic**	Ammonium hydroxide/H_2_O	Basic, buffered at 9.5	NH_4_OH

**Table 2 sensors-23-02596-t002:** Samples obtained with ethylene glycol synthesis: optical and morphological properties.

**Sample**	** $\frac{mol\;\left(H_{2}O\right)}{mol\;\left(Ti{\left(O^{i}Pr\right)}_{4}\right)}$ **	**Normalized** **Raman** **Intensity** **[Count/mW]**	**Peak** **Position** **[cm^−1^]**	**FWHM** **[cm^−1^]**	**Crystallite** **Diameter** **(by Scherrer)** **[nm]**	**NOTE**
**TP23.1**	68.5	85	146.2	-	-	Amorphous
**TP23.2**	75	28	145.0	-	-	Amorphous
**TP23.3**	100	17,505	145.3	17.7	49	Anatase
**TP23.4**	125	140,435	146.0	17.0	48	Anatase
**TP8**	150	127,773	146.7	16.9	29	Anatase
**TP9**	150	180,926	146.8	17.5	37	Anatase
**TP23.5**	150	128,597	145.3	16.1	38	Anatase
**TP26.3**	150	134,703	147.5	17.2	24	Anatase
**TP23.6**	175	122,315	145.5	16.5	26	Anatase
**TP26.4**	346	112,191	145.7	14.7	-	Anatase/Rutile
**TP26.5**	708	1833	143.7	13.7	-	Rutile
**TP26.6**	1431	727	142.6	15.0	-	Rutile

**Table 3 sensors-23-02596-t003:** Fitting parameters and results.

Fitting Parameters		Fitting Results	
$\nu_{0}$	514.5 nm	A	0.957 ± 0.002
$\nu_{m}$	146.8 cm^−1^	R Square	0.90

**Table 4 sensors-23-02596-t004:** Parameters of tested materials: position of the E_g(1)_ peak, crystallite diameter estimated from XRD data, NP dimensions estimated from SEM images, values of A constant and R^2^ of the fitting.

Sample	Peak Position [cm^−1^]	Crystallite Diameter (by Scherrer) [nm]	NP Dimensions (by SEM) [nm]	Instrumental Constant A	R^2^ Fit
**Commercial TiO_2_ [27]**	143	56	~200	0.961 ± 0.006	0.95
**TP9—Glycol**	146.8	42	150 × 300	0.957 ± 0.002	0.90
**TP22.10—Acid**	146.9	13	300 × 700	0.9438 ± 0.0005	0.98
**TP25.1—Acid**	147.2	34	200 × 700	0.960 ± 0.002	0.93
**TP21.3—Basic**	143.9	21	20 × 40	0.959 ± 0.001	0.96
**TP26.9—Basic**	143.7	20	30 ×40	0.9730 ± 0.0004	0.96

## Data Availability

Not applicable.

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
