# Peer review of "A Sol-Gel/Solvothermal Synthetic Approach to Titania Nanoparticles for Raman Thermometry"

_sensors, 2023, doi:10.3390/s23052596_

Round 1

Reviewer 1 Report

The manuscript submitted by Thomas Pretto et al. reports on a combined sol-gel/solvothermal synthesis of titania nanoparticles and their subsequent application as nanothermometers using Raman thermometry.

In my opinion, the paper is well – written and provides an important contribution to the research in the field of materials chemistry, inorganic synthesis and nanothermometry. The language used throughout the text is good, the synthesis seems to have been carried out carefully and the conclusions are well supported by results. I suggest the paper to be published in Sensors after some minor improvements:

·       Line 51: ‘on the order of the diffraction limit’ should probably read ‘in the order of the diffraction limit’

·       Line 57: the sentence ‘To perform Raman nanothermometry is possible to monitor the variation of (…)’ sounds odd, please rewrite.

·       Line 80 and further on throughout the text: the temperature should be written like 500 oC, i. e., with a blank space between 500 and oC

·       Figure 1: nice picture, however, the text next to the graphics is too small and difficult to read. Especially when printed on a paper. Please provide bigger captions.

·       Line 131: what is meant by mol(H2O)/mol(Ti(O)Pr4) = min?

·       Paragraph 3, Figure 2, lines 240/241 + 248: from the comparison between the nanorod dimensions obtained by SEM and results obtained by Scherrer (not Debye-Scherrer !) formula, it is clear that the nanorods are multi-crystalline. Additionally, a particle with a short axis of 300 nm and long axes of 700 nm can hardly be referred to as ‘nanoparticle’. Please comment on that.

·       Line 253: In the sentence ‘peaks 397, 505 and 640 cm-1’, the word ‘at’ seems to be missing: ‘peaks at 397, 505 and 640 cm-1’.

·       Line 401: avoid capital letters in the words ‘Room Temperature’.

·       Line 431: Figure 13 (b) should be written with the capital first letter.

·       Lines 460 + 461: ‘Within each synthesis are reported two different samples made with different hydrolysis ratios: (…)’ should rather read: ‘Within each synthesis, two different samples made with different hydrolysis ratios are reported: (…)’

·       Line 507: ‘with that determined …’ should be replaced by ‘with those determined …’, since the part refers to constants (= plural)

·       References: while the list seems to be comprehensive, I wonder why the last citation is from 2020 (only one paper), while some of the references are even more than 30 years old. In an area which is evolving rapidly, one would expect at least some references from 2021 – 2023.

One last general question: the authors state at least in two places in the manuscript (e.g., Lines 393-394 in Results as well as in Lines 507-508 in Discussion and Conclusions, how their results are in agreement with values obtained using commercially available TiO2 powder. In this case, I wonder where was the main point to carry out the synthesis if the same results could be obtained using commercial TiO2? The authors should add some information to point out the advantages of their as-prepared titania NPs when compared to commercial products.

Author Response

We thank reviewer 1 for his positive comments and suggestions on our paper. All suggested corrections, listed below, have been highlighted in yellow in the text.

We have corrected the following citations. Lines 51, 57, 80, 253, 401, 431 and 507.

We have provided bigger captions for Figure 1.

We have added to the text, line 131, the meaning of 'min', which had been reported only in the SI.

We have corrected, as suggested, lines 240+241 and 460+461.

These references have been added to the text, in order to update them to the state-of-the-art of the last years:   

  • Kondratenko, K.; Hourlier, D.; Vuillaume, D.; Lenfant, S. Nanoscale thermal conductivity of Kapton-derived carbonaceous materials. of Appl. Phys. 2022, 131 (6), 065102. doi.org/10.1063/5.0074407.
  • Kong, N.; Hu, Q.; Wu, Y.; Zhu, X. Lanthanide Luminescent Nanocomposite for Non-Invasive Temperature Monitoring in Vivo. Eur. J. 2022, e202104237. doi.org/10.1002/chem.202104237
  • Nexha, A.; Carvajal, J.J.; Pujol, M. C.; Diaz, F.; Aguilo, M. Lanthanide doped luminescence nanothermometers in the biological windows: strategies and applications. Nanoscale 2021, 13 (17) 7913-7987. doi.org/10.1039/d0nr09150b
  • Liu, S.B.; Yan, L.; Huang, J.S.; Zhang, Q.Y.; Zhou, B. Controlling upconversion in emerging multilayer core-shell nanostructures: from fundamentals to frontier applications. Soc. Rev. 2022, 51 (5), 1729-1765. doi.org/10.1039/d1cs00753j.
  • Jia, M.; Chen, X.; Sun, R.; Wu. D.; Li X.; Shi, Z.; Chen, G.; Shan, C. Lanthanide-based ratiometric luminescence nanothermometry. Nano Res. 2022. https://doi.org/10.1007/s12274-022-4882-7
  • Jin, H.; Yang, M.; Gui, R. Ratiometric upconversion luminescence nanoprobes from construction to sensing, imaging, and phototherapeutics. Nanoscale, 2023,15, 859-906. doi.org/10.1039/D2NR05721B.

We have corrected the text concerning this point:  Additionally, a particle with a short axis of 300 nm and long axes of 700 nm can hardly be referred to as ‘nanoparticle’. Please comment on that.

The definition of a nanoparticle says: 'A nanoparticle is a small particle that ranges between 1 and 100 nanometres in size.' Since our structure is larger than 100 nm, it cannot be defined as NPs, only the smaller ones are NPs.

The responses on the last question, outlined by the referee, are reported in the following and in the main text, and allows us to underline an important point that we have not highlighted well:

“One last general question: the authors state at least in two places in the manuscript (e.g., Lines 393-394 in Results as well as in Lines 507-508 in Discussion and Conclusions, how their results are in agreement with values obtained using commercially available TiO2 powder. In this case, I wonder where was the main point to carry out the synthesis if the same results could be obtained using commercial TiO2? The authors should add some information to point out the advantages of their as-prepared titania NPs when compared to commercial products.”

Synthesis of anatase of different dimensions and morphology opens the way to the possibility of preparing materials with well-defined characteristics, in terms of dimensions, for example, and surface functionalisation, and is the first step for the creation of composite materials. Only by controlling the dimensions and suitably functionalising the external surface of the nanostructures will it be possible to internalise and localise them in biological systems. Moreover, it will open the way to the preparation of composite materials in which the nanothermometer, anatase, can be combined, for example, with a plasmonic material capable of amplifying its thermometric sensitivity and at the same time functioning as a local nanoheater.

Reviewer 2 Report

The paper reports on the synthesis of titania nanoparticles and their use in Raman thermometry in a biologically relevant temperature range. The results obtained are new and interesting; possible applications of the approach proposed include in vivo thermometry of living cells etc. The conclusions are fully supported by the experimental data, the discussion is informative and comprehensive.

I have the following comments:

1. The exact chemical composition of the materials obtained remains unknown. In my opinion, thermal analysis data could supplement the data presented to quantify the content of chemically bound water in titania nanoparticles.

2. Some kind of a control sample (e.g., sol-gel titania annealed at higher temperature, for example at 500-600oC) is required to elucidate the role of bound water in the determination of temperature using Raman thermometry with titania nanoparticles.

3. Some comparison is required between the Raman nanothermometry and well-established luminescent thermometry with non-toxic rare earth compounds. The latter demonstrate very good sensitivity etc.

Author Response

We thank reviewer 2 for comments and suggestions on our paper. The following are the replies to the comments made.

Points 1 and 2. We have characterized the TP9 sample making it a TGA, the result of which is shown in the attached figure. This thermal analysis shows a total mass loss of 5-6% in weight, in the range 30-700°C; it is comparable to different commercial and synthetised TiO2 NPs [Wu, C.Y.; Tu, K-J.; Deng, J-P.; Lo, Y-S.; Wu C-H. Markedly Enhanced Surface Hydroxyl Groups of TiO2 Nanoparticles with Superior Water-Dispersibility for Photocatalysis, Materials 2017, 10, 566.]. In addition, the IR spectra of the samples used for the nanothermometry measurements were also recorded and reported in the figure. From the comparison of the O-H bond stretching signal of TP9 sample, in the 2500-3500 cm-1, and TP21.3, TP22.10, TP25.1, and TP26.9 IR signals, it is possible to affirm that all samples possess a comparable quantity of water, which is also comparable with commercial anatase, where the physisorbed and chemisorbed water content is investigated [Li, G.; Li, G.; Boerio-Goates, J.; Woodfield, B.F. High Purity Anatase TiO2 Nanocrystals: Near Room-Temperature Synthesis, Grain Growth Kinetics, and Surface Hydration Chemistry. JACS. 2005, 127 (24) 8659].

It is also interesting to outline that the performances of materials such as nanothermometers are independent of the water content. The choice to monitor a Raman mode, in particular the ratio between signal aS and S, provides a sort of internal normalization to each single measurement making the signal and the methodology, used to estimate the temperature, dependent on the Eg(1) vibrational mode of anatase and independent of any water content. Moreover, the materials will be properly functionalised for use in biological media, such as water-based solutions, and also in this case the presence of a residual water in the pristine material will not influence the temperature measurement.

Taking into account point 3, the comparison with fluorescent thermometer materials is interesting. The strength of Raman nanothermometers lies in the possibility of being able to use them in a wider range of wavelengths than fluorescent materials which are excited only at well-defined wavelengths.

The observation on the useful wider wavelength range has been added in the text and highlighted in yellow.

Reviewer 3 Report

In the manuscript, titania nanoparticles (NPs) have been evaluated as a nanothermometer material that utilizes the Raman technique to determine local temperature non-invasively. Biocompatible TiO2 NPs were produced using a combination of sol-gel and solvothermal green synthesis techniques with the aim of achieving pure anatase samples. The optimization of three synthesis protocols has allowed for the production of materials with well-defined crystallite dimensions, favorable morphologies, and enhanced dispersibility. Characterization was carried out through XRD and room-temperature Raman measurements, which confirmed the synthesized samples as single-phase anatase TiO2, and SEM measurements showed the nanometric dimensions of the NPs. Stokes and anti-Stokes Raman measurements were collected in the temperature range of 293-323 K, with the excitation laser operating at 514.5 nm, for a range of biological applications. The laser power was chosen with care to avoid any heating effects from laser irradiation. I recommend the acceptance of the paper with grammatic errors revised, e.g., in line 485, the word ‘synthesys’ was misspelled.

Author Response

We thank reviewer 3 for his positive comments and suggestions on our paper. The grammatic errors have been corrected and highlighted in yellow in the text.